# Thrombotic events and rebleeding after hemorrhage in patients taking direct oral anticoagulants for non-valvular atrial fibrillation

Daisuke Yanagisawa[1], Koichiro Abe[1]*, Hirohito Amano[1], Shogo Komatsuda[1], Taku Honda[1], Daisuke Manabe[1], Hirosada Yamamoto[2], Ken Kozuma[2], Shinya Kodashima[1], Yoshinari Asaoka[1], Takatsugu Yamamoto[1], Atsushi Tanaka[1]

1 Division of Gastroenterology, Department of Medicine, Teikyo University School of Medicine, Tokyo, Japan, 2 Division of Cardiology, Department of Medicine, Teikyo University School of Medicine, Tokyo, Japan

* abe@med.teikyo-u.ac.jp

**Data Availability Statement:** All relevant data are within the paper.

## Abstract

Several direct oral anticoagulants have been developed to prevent cardiogenic thrombosis in patients with atrial fibrillation, on the other hand, have the complication of bleeding. Since clinical course after bleeding with direct oral anticoagulant remains unclear, the present retrospective cohort study was to clarify the course after hemorrhage among patients receiving direct oral anticoagulants. Among all 2005 patients prescribed dabigatran, rivaroxaban, apixaban, or edoxaban between April 2011 and June 2017, subjects comprised 96 patients with non-valvular atrial fibrillation who experienced relevant bleeding during direct oral anticoagulant therapy (Bleeding Academic Research Consortium type 2 or above). The clinical course after hemorrhage was reviewed to examine whether rebleeding or thrombotic events occurred up to the end of December 2019. Gastrointestinal bleeding was the most frequent cause of initial bleeding (57 patients, 59%). Rebleeding occurred in 11 patients (4.5%/year), with gastrointestinal bleeding in 10 and subarachnoid hemorrhage in 1. All rebleeding occurred in patients who resumed anticoagulation therapy. Another significant factor related with rebleeding included past history of gastrointestinal bleeding. On the other hand, major adverse cardiac and cerebrovascular events occurred in 6 patients older than 75 years old or more (2.5%/year), with systemic thrombosis in 4 and cardiac death in 2. All 4 patients with systemic thrombosis withheld anticoagulants after index bleeding, although only 10 patients withheld anticoagulation therapy. Rebleeding should be taken care of when anticoagulants are resumed after bleeding, particularly among patients who initially experienced gastrointestinal bleeding. Systemic thrombosis occurred at a high rate when anticoagulant therapy was withheld after bleeding.

**Funding:** The authors received no specific funding for this work.

**Competing interests:** The authors have declared that no competing interests exist.

## Introduction

With the continued rapid aging of society, the number of patients suffering from atrial fibrillation (AF) has increased in developed countries [1, 2]. Several direct oral anticoagulants (DOACs) have recently emerged for the prevention of thrombosis in patients with AF. DOACs have characteristics of improved adherence, easier handling and effectiveness comparable to that of vitamin K antagonists, whereas a drawback could be a higher risk of bleeding especially in the gastrointestinal tract, a common site of bleeding during antithrombotic therapy. The incidence of major gastrointestinal bleeding (GIB) among patients taking DOACs has been reported as 1–3% in randomized controlled trials [3–7]. Previous studies have indicated an increased incidence of GIB among AF patients taking dabigatran and rivaroxaban compared with those taking warfarin [8–12].

In addition, continuation of warfarin is considered important for preventing thrombotic adverse events, even after serious bleeding developed. Restarting warfarin after major GIB is reportedly associated with significant reductions in thromboembolism and all-cause mortality [13–15]. Meanwhile, anticoagulant interruption is known to be related to increased risks of all-cause mortality and thrombosis, but no decrease in risk of major bleeding [16, 17]. However, information on the clinical course after bleeding during DOAC therapy remains limited, and both the incidences of rebleeding and thrombosis and factors associated with the development of such adverse events are uncertain [18]. The present study was to clarify the occurrence of rebleeding and thrombotic events in post-bleeding patients and the relation with clinical factors such as resumption or discontinuation of anticoagulation therapy.

## Materials and methods

### Study subjects

This was a single-center, retrospective cohort study. Study subjects were selected from patients at Teikyo University Hospital in Japan. The patients who had been prescribed a DOAC in the form of dabigatran, rivaroxaban, apixaban, or edoxaban between April 2011 and June 2017 were identified from prescription lists. From them, the patients who had been prescribed DOAC for non-valvular atrial fibrillation (NVAF) and experienced clinically relevant bleeding (Bleeding Academic Research Consortium (BARC) type 2–5) [19] during DOAC therapy were included. The exclusion criteria were involved prescribing DOAC except non-valvular atrial fibrillation, inpatients, less than 1 month from prescribed DOAC, unable to follow up after bleeding and bleeding caused by medical procedures.

BARC proposes 5 bleeding types. Type 0 is no bleeding. Type 1 is bleeding that is not actionable and does not cause the patient to seek medical attention. Type 2 bleeding includes any clinically overt sign of hemorrhage that is actionable and requires diagnostic studies, hospitalization, or treatment by a healthcare professional. Type 3 bleeding is divided into 3 categories. Type 3a bleeding includes any transfusion with overt bleeding plus a hemoglobin drop of 3 to < 5 g/dL (provided the hemoglobin drop is related to bleeding). Type 3b bleeding includes overt bleeding plus a hemoglobin drop of $\geq$ 5 g/dL (provided the hemoglobin drop is related to bleeding), cardiac tamponade, bleeding requiring surgical intervention for control (excluding dental/nasal/skin/hemorrhoid), and bleeding requiring intravenous vasoactive agents. Type 3c bleeding includes intracranial hemorrhage and intraocular bleeding compromising vision. Type 4 bleeding is associated with procedures of coronary artery bypass grafting, such as perioperative intracranial bleeding within 48 h and reoperation after closure of sternotomy for the purpose of controlling bleeding. Type 5 bleeding is fatal.

## Data collection

The medical records of subjects were reviewed to identify whether bleeding or major adverse cardiac and cerebrovascular events (MACCE), including systemic thrombosis, myocardial infarction and cardiac death, had developed up to the end of December 2019 (primary endpoint). Observations ceased when bleeding or MACCE occurred, or the patient stopped visiting our institution for >6 consecutive months without a documented reason (dropout cases). Baseline characteristics of subjects, meaning data at the time of the index bleeding event, were also investigated, including biographic data (age, sex, height, weight), type of DOAC, comorbidities (hypertension, dyslipidemia, diabetes mellitus, chronic heart failure, ischemic heart disease, cerebrovascular disease, peripheral artery disease, chronic kidney disease, chronic obstructive pulmonary disease, liver cirrhosis, advanced malignant diseases), and concomitant medications (steroids, nonsteroidal anti-inflammatory drugs, low-dose aspirin, adenosine diphosphate receptor P2Y12 antagonists, or proton pump inhibitors (PPIs)). CHADS2 score and HAS-BLED score were also obtained from the background data.

## Statistics

All statistical analyses were performed using SPSS Statistics version 24 (IBM Japan, Tokyo, Japan). Continuous variables are presented as median (interquartile range [IQR]) and categorical variables were shown by number and/or percentage. In order to minimize effects of confounders, differences in ratios between groups were evaluated by linear correlation model using inverse probability weighting methods. Differences with two-sided alpha levels of <0.05 were determined as statistically significant.

## Ethics

This protocol was approved by the institutional review board of Teikyo University prior to the study (approval number TU-18-216). All methods were carried out in accordance with relevant guidelines and regulations. The need to obtain informed consent was waived by the ethics committee that approved the study, given the retrospective design of the study.

## Results

### Characteristics of subjects at index bleeding

Fig 1 depicted the details of study flow. First of all, 2005 patients who had been prescribed DOAC were obtained. There were 172 patients prescribed DOAC for NVAF and experienced clinically relevant bleeding. Among those, 76 patients were excluded due to prescribed only in the hospital (n = 48), prescribed less than 1 month (n = 12), unable to follow up after bleeding (n = 10) and bleeding after medical procedures (n = 6).

Table 1 shows the characteristics of subjects, and Table 2 shows the site and severity of bleeding. As shown in Table 1, 96 patients were eligible to the subjects. The median age was 76 years old, and male dominant. Prescribed DOACs were rivaroxaban in 51 (53%), dabigatran in 22 (23%), apixaban in 18 (19%), and edoxaban in 5 (5%). The median CHADS2 score was 2, and HAS-BLED score was 3. Regarding comorbidities, hypertension, dyslipidemia, chronic heart failure and chronic kidney disease coexisted with high rate more than 50%. As for concomitant drugs, low-dose aspirin was given in 27% and P2Y12 was in 17%. The rate of concomitant prescription of proton pump inhibitors reached to 51%.

Table 2 shows that 57 of all the subjects had experienced GIB (59%), with lower GIB representing the majority, while upper GIB had a higher rate of BARC type 3 or more, meaning

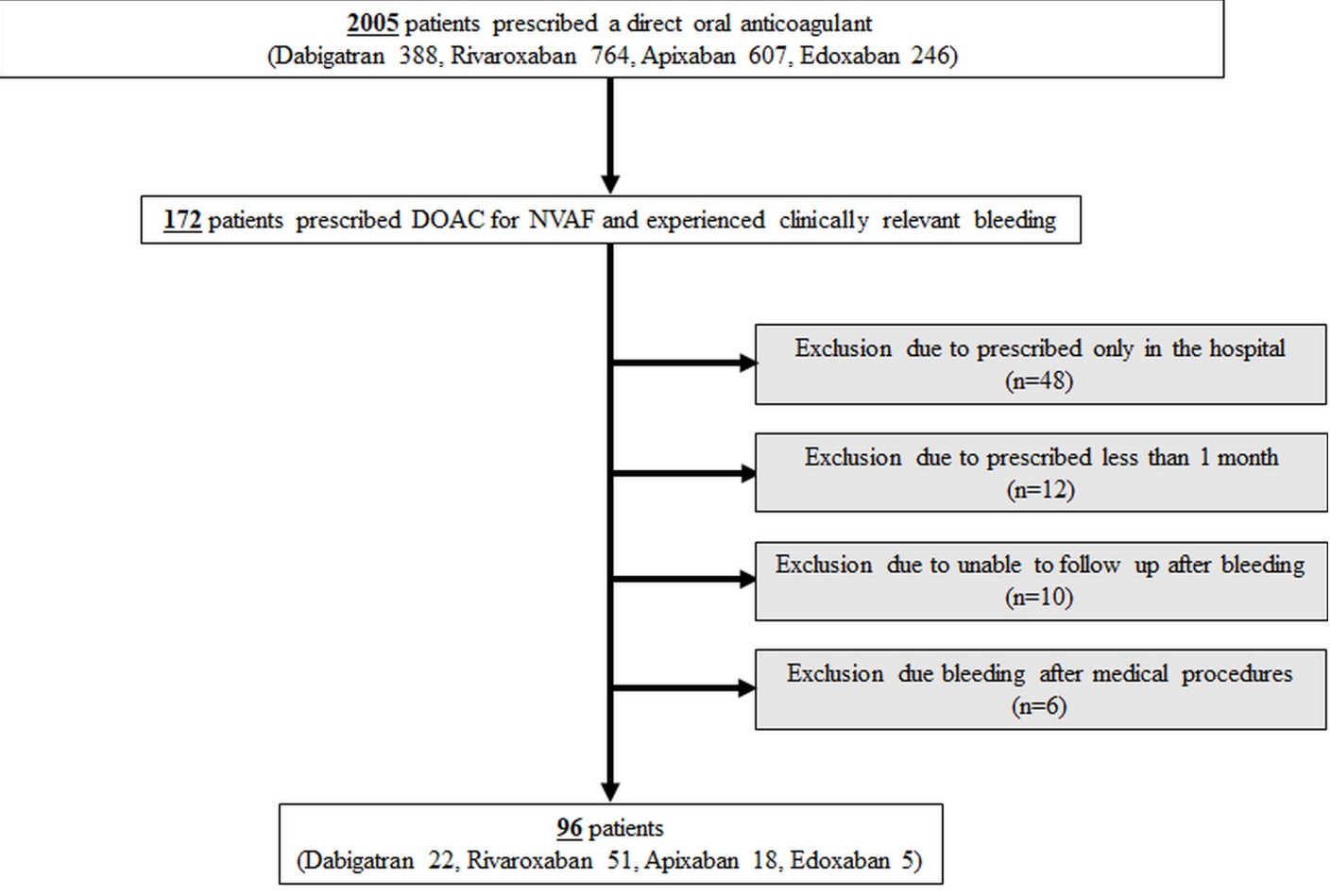

**Fig 1. Flowchart for selection of study subjects.** Abbreviations: DOAC, direct oral anticoagulant; NVAF, non-valvular atrial fibrillation.

that upper GIB tended to be more serious. The major cause of upper GIB was peptic ulcer, and reasons for lower GIB included diverticulum, hemorrhoids, malignancy, and vascular ectasia. Other sites of bleeding included cutaneous, ocular, nasal, oral, urological, and intracranial bleeding. Most episodes of bleeding remained mild in severity, excluding intracranial bleeding.

## Rebleeding and thrombosis after initial bleeding

Table 3 summarizes data concerning rebleeding and thrombotic events after initial bleeding in subjects. The observation period was 242.6 patient-years. Of all 96 patients, 11 developed rebleeding (4.5%/year), including GIB in 10 and subarachnoid hemorrhage in 1. The lower GI tract was the majority site of rebleeding, and the most common cause was colonic diverticulum. On the other hand, MACCE occurred in 6 patients (2.5%/year), with systemic thrombosis in 4 (2 death) and cardiac death in 2. There was no case of cardiac infarction.

Table 4 describes status of anticoagulation therapy after initial bleeding. Most patients had resumed anticoagulation therapy after bleeding, excluding 10 patients for whom prescription of all oral anticoagulants was withheld. About half of the subjects resumed the same drug with the same dosage as before initial bleeding, while 40% of them changed

**Table 1. Characteristics of study subjects at index bleeding.**

| Characteristics | Total |
|---|---|
| Number of patients | 96 |
| Age (years), median (IQR) | 76 (70–81) |
| Sex (male), n (%) | 57 (59.3) |
| Height (m), median (IQR) | 1.59 (1.50–1.67) |
| Weight (kg), median (IQR) | 58.3 (48.6–67.0) |
| CHADS2 score, median (IQR) | 2 (2–3) |
| HAS-BLED score, median (IQR) | 3 (3–4) |
| DOAC before bleeding, n (%) | |
| Dabigatran | 22 (22.9) |
| Rivaroxaban | 51 (53.1) |
| Apixaban | 18 (18.8) |
| Edoxaban | 5 (5.2) |
| Comorbidities, n (%) | |
| Hypertension | 82 (85.4) |
| Diabetes mellitus | 22 (229) |
| Dyslipidemia | 56 (58.3) |
| Chronic heart failure | 60 (62.5) |
| Ischemic heart disease | 27 (28.1) |
| Cerebrovascular disease | 13 (13.5) |
| Peripheral arterial disease | 6 (6.3) |
| Chronic obstructive pulmonary disease | 8 (8.3) |
| Liver cirrhosis | 0 (0.0) |
| Advanced malignancy | 13 (13.5) |
| Chronic kidney disease | 63 (65.6) |
| Medications, n (%) | |
| Low-dose aspirin | 27 (28.1) |
| P2Y12 | 16 (16.7) |
| Nonsteroidal anti-inflammatory drugs | 8 (8.3) |
| Steroids | 4 (4.2) |
| Proton pump inhibitor | 51 (52.1) |

IQR: interquartile range, n: number, DOAC: direct oral anticoagulants,* P2Y12: adenosine 2 phosphate receptor P2Y12 antagonist.

drugs to other OAC. The duration of withdrawal was less than 14 days in 90% of patients resuming anticoagulant therapy.

Table 5 shows the impact of factors on development of rebleeding. Rate of resumption of anticoagulation therapy was 100% in patients experienced rebleeding. Past history of GIB was a significant factor predicting the development of rebleeding. Concomitant prescription of antiplatelet was not significantly related to rebleeding. In patients with rebleeding, HAS-BLED scores tended to be higher than those in patients without rebleeding, although it remained statistically insignificant.

Table 6 describes the occurrence of MACCE and possible relating factors. All 6 patients suffering from MACCE were 75 years of age or more. Anticoagulation therapy was withheld in 4 of 6 patients experiencing MACCE. Especially, all 4 patients with systemic thrombosis did not resume anticoagulants after initial bleeding events.

**Table 2. Cause and severity of index bleeding in study subjects.**

| Site of bleeding | Number of patients | Severity of bleeding (BARC type 2/type 3 or more) |
|---|---|---|
| Total | 96 | 64/32 |
| **Gastrointestinal tract** | 57 | **31/26** |
| Upper | 17 | 6/11 |
| Peptic ulcer | 12 | 3/9 |
| Gastric polyp | 2 | 1/1 |
| Gastric cancer | 2 | 1/1 |
| Vascular ectasia | 1 | 1/0 |
| Lower | 40 | 25/15 |
| Diverticulum | 14 | 7/7 |
| Vascular ectasia | 9 | 6/3 |
| Hemorrhoid | 8 | 6/2 |
| Cancer | 4 | 2/2 |
| Inflammation | 4 | 3/1 |
| Rectal ulcer | 1 | 1/0 |
| **Other site** | 39 | **33/6** |
| Intracranial | 5 | 0/5 |
| Nasal cavity | 9 | 9/0 |
| Urinary tract | 7 | 7/0 |
| Oral cavity | 6 | 6/0 |
| Cutaneous/Subcutaneous | 7 | 7/0 |
| Ocular region | 4 | 3/1 |
| Joint | 1 | 1/0 |

Abbreviation: BARC, criteria of the Bleeding Academic Research Consortium.

## Discussion

The present results showed that the gastrointestinal tract was the most frequent site of index bleeding, and GIB tended to be more serious than bleeding from other sites. HAS-BLED score, which represents factors associated with bleeding during warfarin treatment, indicates that a past history of bleeding is an obvious risk for future bleeding [20–24]. Our previous study indicated that a history of GIB was one of the significant risk factors for rebleeding from gastrointestinal tract among patients who have received DOACs [25]. Since the rebleeding rate was higher than the incidence of MACCE, the risk of rebleeding should be kept in mind, especially in the gastrointestinal tract.

   Concerning the site of GIB, a high proportion of cases showed sites other than the upper gastrointestinal tract. Previous studies have reported that simultaneous PPI with DOAC has prophylactic effects against the onset of upper GIB [26–30]. In the present study, the upper GIB would have been suppressed probably because the rate of concomitant PPI use was high, in more than half of patients. On the other hand, no solution to prevent bleeding has been established for other parts of the gastrointestinal tract. Lanas et al. mentioned that concomitant use of PPIs among patients on antithrombotic drugs was associated with a reduced risk of upper GIB, but not a lower risk of GIB [30]. Although the severity of lower GIB tended to be lower than that of upper GIB, as shown in Table 2, lower GIB sometimes becomes life-threatening because of the risk of interruption to pharmacotherapy. Lower GIB warrants caution, even if these symptoms and signs are not serious.

**Table 3. Rebleeding and thrombotic events after initial bleeding in patients on direct oral anticoagulant therapy.**

| Data | |
|---|---|
| Observation data | |
| Number of patients | 96 |
| Period, patient-years | 242.6 |
| median (IQR) | 2.3 (0.8–3.7) |
| Dropout, n (%) | 3 (3.1) |
| Adverse events; n (%/year) | |
| Rebleeding | 11 (4.5) |
| Gastrointestinal | 10 (4.1) |
| Gastric vascular ectasia | 1 |
| Colonic diverticula | 7 |
| Colonic cancer | 1 |
| Hemorrhoid | 1 |
| Intracranial | 1 (0.4) |
| Subarachnoid hemorrhage | 1 |
| Major adverse cardiac and cerebrovascular event | 6 (2.5) |
| Cardiac death | 2 (0.8) |
| Myocardial infarction | 0 (0) |
| Systemic thrombosis | 4 (1.6) |

Abbreviations: IQR, interquartile range; n, number.

For the resumption of anticoagulant therapy after bleeding, a previous study indicated no increase in the risk of embolism development if warfarin is resumed within 30 days after bleeding, while restarting warfarin within 7 days was associated with an increased risk of recurrent GIB compared to restarting after 30 days [16]. As for DOAC, Sengupta et al. reported that in 1338 patients with GIB during DOAC therapy, restarting DOAC therapy within 30 days after admission due to GIB was not associated with thromboembolism or

**Table 4. Status of anticoagulation therapy after initial bleeding during direct oral anticoagulants in patient with nonvalvular atrial fibrillation.**

| Anticoagulation therapy after index bleeding, n (%) | |
|---|---|
| Restart with same drug on same dose | 46 (47.9) |
| Restart with same drug on reduced dose | 3 (3.1) |
| Restart with other drugs | 37 (38.5) |
| To dabigatran | 2 |
| To rivaroxaban | 3 |
| To apixaban | 16 |
| To edoxaban | 4 |
| To warfarin | 12 |
| Withhold medication | 10 (10.4) |
| Total | 96 (100) |
| Duration from bleeding to resume therapy (n = 86) | |
| ≤ 7 days | 69 (80.2) |
| 8–14 days | 9 (10.5) |
| 15–30 days | 5 (5.8) |
| ≥ 31 days | 3 (3.5) |

**Table 5. Relationship between occurrence of rebleeding and major factors in patients with non-valvular atrial fibrillation.**

| Characteristics | Rebleeding (+) | Rebleeding (-) | Adjusted hazard ratio* (95% Confidence interval) | P-value |
|---|---|---|---|---|
| Number of patients | 11 | 85 | - | - |
| Time to rebleeding, month (range) | 18 (1–60) | - | - | - |
| **Past GIB, n (%)** | 10 (91.0) | 53 (62.3) | 11.672 (1.390–98.009) | **0.024** |
| **Resumption of anticoagulant, n (%)** | 11 (100) | 75 (88.2) | $9.6 \times 10^8$ ($3.7 \times 10^8$-$24.1 \times 10^8$) | **<0.001** |
| Concomitant antiplatelet, n (%) | 6 (54.5) | 25 (29.4) | 0.925 (0.216–3.961) | **0.916** |
| HAS-BLED score, median (IQR) | 4 (3.5–4.5) | 3 (3–4) | 1.672 (0.924–3.023) | 0.089 |

* Evaluated by generalized linear model using propensity score weighting by inverse probability weighting method. Abbreviations: n, number; GIB, gastrointestinal bleeding, IQR, interquartile range.

recurrent GIB within either 90 days or 6 months [18]. In the present study, the resumption rate within 30 days was approximately 90%. However, although only 10 patients were still withheld DOACs after initial bleeding, MACCE occurred in 4. Especially, all 4 patients with systemic thrombosis withheld anticoagulants. In the literature, the strongest predictors for withholding anticoagulants were concomitant antiplatelet agents and a history of bleeding, and 40–50% of patients with AF are regarded as being in a state of anticoagulant under-use [22, 31–33]. On the other hand, it is to be noted that all cases of rebleeding resumed anticoagulation therapy, and more than half of them were also on antiplatelet drugs concurrently. The necessity for combined use of antithrombotic drugs may be considered carefully in patients with a history of bleeding.

Limitations of this study included the use of a single facility, the small number of subjects, and the retrospective study design. In retrospective studies, the number of dropouts is often problematic, but was only 4 patients in this study, and was thus considered to have had little effect on the results. Research involving multiple facilities is desirable in the future.

## Conclusions

The present study showed that GIB was common and serious among patients taking DOACs. The incidence of rebleeding seemed high, while withholding of anticoagulants was associated with a high rate of developing thromboembolism.

**Table 6. Relationship between occurrence of MACCE after initial bleeding and major factors in patients with non-valvular atrial fibrillation.**

| Characteristics | MACCE (+) | MACCE (-) | Adjusted hazard ratio* (95% Confidence Interval) | P-value |
|---|---|---|---|---|
| Number of patients | 6 | 90 | - | - |
| Time to MACCE, month (range) | 8.5 (5–23) | - | - | - |
| **75 years old or more, n (%)** | 6 | 48 | $7.0 \times 10^8$ | **<0.001** |
| | (100%) | (53.3%) | ($2.6 \times 10^7$-$1.8 \times 10^{10}$) | |
| Sex (male), n (%) | 2 | 61 | 0.132 | 0.057 |
| | (33.3) | (63.5) | (0.016–1.064) | |
| Withhold of anticoagulant, n (%) | 4 (66.7) | 6 (6.7) | 3.243 | 0.290 |
| | | | (0.367–28.624) | |
| CHADS2 score, Median (IQR) | 2.5 (2–3) | 3 (2–3) | 0.985 | 0.965 |
| | | | (0.499–1.446) | |

*: generalized linear model using propensity score weighting by inverse probability weighting method. Abbreviations: MACCE, major adverse cardiac and cerebrovascular events; IQR, interquartile range.

## Author Contributions

**Conceptualization:** Takatsugu Yamamoto.

**Data curation:** Daisuke Yanagisawa, Hirohito Amano, Shogo Komatsuda, Taku Honda, Daisuke Manabe, Hirosada Yamamoto, Takatsugu Yamamoto.

**Formal analysis:** Koichiro Abe, Shinya Kodashima, Yoshinari Asaoka, Takatsugu Yamamoto.

**Supervision:** Ken Kozuma, Atsushi Tanaka.

**Validation:** Shinya Kodashima, Yoshinari Asaoka.

**Writing – original draft:** Daisuke Yanagisawa.

**Writing – review & editing:** Koichiro Abe, Atsushi Tanaka.

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
