## [Decision Letter · Decision Letter 0]

22 Feb 2021

PONE-D-20-38582

Clinical course after hemorrhage in patients taking direct oral anticoagulants

PLOS ONE

Dear Dr. Abe,

Thank you for submitting your manuscript to PLOS ONE. After careful consideration, we invite you to submit a revised version of the manuscript that addresses the points raised during the review process.

Your paper was reviewed by four experts in the field. All reviewers are concerned by the confusing study design and data presentation, and incomplete data analyses to some extents. Please read the comments carefully, and address the issues accordingly. Although novelty is not considered for judge by the journal's policy, please explain what this study can add to our current knowledge in clinical practice. Since extensive improvement is required, please contact the editorial office for extension of deadline if needed.

We look forward to receiving your revised manuscript.

Kind regards,

Tomohiko Ai, M.D., Ph.D.

Academic Editor

PLOS ONE

2. Please upload a new copy of Figure 2 as the detail is not clear. Please follow the link for more information: https://blogs.plos.org/plos/2019/06/looking-good-tips-for-creating-your-plos-figures-graphics/" https://blogs.plos.org/plos/2019/06/looking-good-tips-for-creating-your-plos-figures-graphics/ No Figure

Reviewers' comments:

Reviewer's Responses to Questions

**Comments to the Author**

1. Is the manuscript technically sound, and do the data support the conclusions?

Reviewer #1: Partly

Reviewer #2: Yes

Reviewer #3: Partly

Reviewer #4: No

2. Has the statistical analysis been performed appropriately and rigorously? 

Reviewer #1: Yes

Reviewer #2: No

Reviewer #3: Yes

Reviewer #4: I Don't Know

3. Have the authors made all data underlying the findings in their manuscript fully available?

Reviewer #1: Yes

Reviewer #2: Yes

Reviewer #3: No

Reviewer #4: Yes

4. Is the manuscript presented in an intelligible fashion and written in standard English?

Reviewer #1: Yes

Reviewer #2: Yes

Reviewer #3: Yes

Reviewer #4: Yes

5. Review Comments to the Author

Reviewer #1: In this manuscript, Yanagisawa et al describe the bleeding and ischemic events in a cohort of DOAC-treated patients after a first hemorrhage. It is a retrospective, descriptive, chart review study that is nonetheless clearly written and informative. It is well curated and attention has been paid to the details. I have (very few) suggestions touching upon a number of structural issues.

Major comments

-This is a descriptive study or patients who have already bled. It cannot speak to incidence of first bleeding, risk factors for bleeding etc. The authors are generally cautious about this throughout the manuscript yet in the final conclusions the state that “the present study showed that GIB was common and serious among patients taking DOACs,” which would require knowing the incidence of bleeding among the 2005 patients mentioned in the methods section that were prescribed DOAC during this period. They then go on to state that “clinicians should endeavor to…” this study cannot speak to what should be done in practice. It is clear that patients who do not resume anticoagulation are generally of greater hemorrhagic and thrombotic risk than those who do not resume and that therefore, retrospective data cannot resolve the question over what clinicians should or should not do when facing a patient who had a major bleeding. Indeed, precisely because this is the case, a number of trials are ongoing randomizing resuming vs. not resuming anticoagulation in patients with intracranial hemorrhage (I am less sure about those with gastrointestinal bleed, but the question over confounders stands). While I am sympathetic to the idea that resuming should be the default position based on the best data we have, this study cannot be used to advise clinicians on what to do.

Minor concerns/suggestions:

-A distinction between spontaneous and traumatic and surgical bleedings should be made throughout the manuscript. At least some are procedure-related based on the table (polypectomy). Nasal bleeding may be related to ENT examinations etc etc. I would judge these patients to have a lower re-bleeding risk (other than related to further procedures) and would rarely hesitate to recommend re-starting anticoagulation; at least it would give me less pause than a spontaneous major bleed would. This is important data.

-I have some misgivings about splitting the cohort into GI bleeding and “other.” I worry that this was a post-hoc split the authors may have made after finding a statistical difference in severity or to have a “comparison” in what is really a descriptive study. The group of other is very heterogeneous to the extent that it becomes a meaningless category (what do nasal bleedings have to do with intracranial hemorrhages). I suggest focusing on the description of all hemorrhages, rather than on the comparison between these two groups, which should just be a side issue. In line with this, it is very hard to know what to do with figure 2, and I suggest removing it. Comparing these two groups, which are otherwise not comparable, is not useful. An informative comparison between these two very different groups would require taking into account many variables, both known and unknown, and this cannot be done in this small patient cohort. A crude comparison between patients who bleed from the GI tract and those who bleed from other sites can be deceiving

Reviewer #2: The authors have reported outcomes in a cohort of patients taking DOACs. The data contained in the paper are not particularly bovel. There are trials and metaanlaysis already reporting bleeding rates in patients taking DOACs but the autjors have added some mroe "real world" data to the body of the literature. Although the overall size of the group is small, once getting down to the subgroups, it is unteresting to see the outcomes in those following reintroduction of anticoagulant after a bleeding episode. Overall the data once again confirm that decsions about reintroduction of anticoagulation need to balance the risk of rebleeding, troublemsome but rarely life-altering against the risk of thromboembolism which can be more clinicaly signficant. The equation usually falls in favour of restarting anticoagulation and this study provides some real workd data to support this. Overall the manuscipt is adequately structured and presented. The writing is suficiently concise. There are a few areas that would benefit from further clarification.

1. In the abstract, it would be very informative to actually give the numerical data for rebleeding and thrombosis speparately for those that had anticoagulation resumed and those that did not. As currently writen it is not clear what these figures for complications after the initial bleeding episode represent.

2. The flow chart explaining subject flow through the study needs more clarity. The reasons for exclusion need to be given and the disposition of subjects made clearer. Simialrly from over 2000 patients with atrial fibrillation there appear to be no deaths during the follow-up period. Is this really the case?

3. Although the authors have usedthe data they have available. The paper should contain a formal statement about samoke size and power calculation related to the planned primary outcome, this will enable the reader to determine if the study is really able to answer the question that the authors have raised. The methods section needs to be clearer about what the primary hypothesis being examined in this study is.

4. The discussion about subjects excluded from the study, currently placed in data collection, more appropriatelyly belongs in the study design and subjects section.

5. Although the focus of this study is on outcomes after the 1st bleeding episode, it is unfortunate given the comprehensive follow up that Table 1 does not include the data for the group that had no bleeding, to allow clearer comparsion.

6. The data of most interest to clinicians will be the outcomes after the 1st bleed, and the differenttiation between those restarted and those without anticoagulation. Hence burying the combined data at the bottom of a summary table 3 is rather unhelpful. These data should be represented separately, although the numbers are small.

7. The authors have provided a lot of data on the co-morbidities of the group post-index bleeding. I am not sure such granualr detail helps at all, especially as the authors have not clearly defined what they consider the inclusions for each of their criteria. Including the data on HAS-BLED scores for the rebleeding and non-rebleeding groups is essntial. This would allow greater generarisation.

8. The overall rebleeding rate seems comparable to other studues, again the authors should be able to estimate from HAS-BLED scores whether the rebleeding rate in their cohort is higer or lower than anticipiated.

9. The authors seem rather over reliant in "p" values to assess differences between groups. It would be more appropriate to repesent these differences a odds ratio/hazaed ratio with 95% confidence intervals. Hence tables 5 and 6 should have these data included For clarity, it would be reasonable to leave out the OR/HR where there is clearly no difference but where the paper cites a "p" value the OR/HR with confidence interval is the correct way to represent this.

Reviewer #3: Comments to authors

This is a neat descriptive study of patients on NOAC who experience bleeding.

There are three points where we need more detailed information.

1. It would be very useful to present more details as to how the anticoagulant drug was restarted. In Table 3 we are informed that 45 pts restarted same drug and 45 changed medication. I believe that this is very vague. We should know if the dose was reduced, if the NOAC was changed (and also from which agent to which agent the change was made) or if warfarin was recommended. A table with the number of pts who remained on same NOAC dose, who restarted lower dose of NOAC and who switched NOAC (or went back to warfarin) would do.

2. A large number of pts was also receiving concomitantly antiplatelet drugs. And it appears that AFTER the bleeding episode many (or all?) continue on them. For example 28 pts were on aspirin (and 17 on P2Y12) before bleeding (table 1) and, surprisingly, 28 remain on aspirin (and 17 on P2Y12) after bleeding (table 5 and 6). Similarly the same number of pts is on PPI before and after bleeding (55 pts) which is odd. The common practice in AF pts who bleed is to withdraw the antiplatelet therapy first rather than stop the NOAC.

3. The rate of MACCE in this group of 102 pts is 2.5%/year. It would be interesting to have the rate for the 2005 pts overall (to confirm that pts who bleed are a high risk group)

4. Accordingly the above issues should be commented in the discussion.

Reviewer #4: The authors report on a cohort of patients with DOAC-related bleeding at a single hospital site in Japan. Previous literature in this area is based on similar observational cohorts at high risk of bias. Some comments are included below for consideration prior to publication.

Introduction

- DOACs have been approved for clinical use for over 10 years (i.e. no longer "recent"). In light of this, what additional information does this study provide beyond existing literature?

- Anticoagulant discontinuation after bleeding is associated with a higher risk of thrombosis and death, but studies were limited by baseline confounding and likely differences in prognosis among patients who resumed versus those who did not resume anticoagulation after bleeding

- it would be helpful to provide specific objectives for readers. As written, "clarify the course after bleeding" is vague.

Methods

- In general, additional details regarding the methods are required.

- Further, some results are included in the methods section (i.e. 2005 patients)

- Please list specific inclusion/exclusion criteria at the beginning of the Study Population section. Patients with AFib? Any dose of DOAC? Why were patients receiving DOACs for less than one month excluded?

- Were these patients identified by outpatient prescriptions?

- Did patients have both reliable inpatient and outpatient follow-up?

- What does "able to be followed up" mean? This indicates that they may not have captured all potential patients. What about patients who weren't able to be followed? It would be important to document loss to follow-up.

- Please clarify the reason BARC was selected as opposed to ISTH definitions which are commonly used in this population

- What does "clinical course was examined every month" mean?

- Did you collect information about DOAC dosing?

- No adjustment for baseline confounding was done (e.g. by regression). Some justification should be provided about why associations were not explored using regression analysis.

- What was the duration of follow-up?

Results

- the rate of clinically relevant bleeding in this cohort seems low (102 out of 2005 patients) over 6 years

- Table 1 should indicate the type of bleeding at cohort entry. Not sure why GI bleeding and "Other bleeding" were separated out into columns. Please include individual DOACs on separate lines (as opposed to D/R/A/E)

- What is meant by "upper GIB tended to be more serious"? Hospitalization? ICU admission? Hemodynamic instability?

- "Tendency to become serious" is vague

- "Little difference was seen between patients with GIB and those with other bleeding" with respect to comorbidities is also vague

- Any differences in dose for restarting anticoagulation?

- Did severity of bleeding or site of bleeding influence therapy?

- Did you capture all-cause death within your cohort? HOw was cardiac death adjudicated?

- When did re-bleeding occur?

- Remove comorbidites from Tables 5 and 6

- What about loss to follow-up?

Discussion

- the conclusions are overstated. It is not clear how the authors showed conclusively that resumption of anticoagulation was not associated with rebleeding as it is stated and based on the analyses conducted

- what does this study add to existing literature?

- The COMPASS trial showed that prophylactic PPI did not influence bleeding (factorial randomization)

- Anticoagulant related GI bleeding is more than "troublesome" with mortality rates of ~10% at 30 days in some cohorts

- Clinical recommendations about the timing of treatment re-initiation is beyond the scope of this paper and remains highly uncertain

6. PLOS authors have the option to publish the peer review history of their article (what does this mean?). If published, this will include your full peer review and any attached files.

Reviewer #1: No

Reviewer #2: **Yes: **Ian L. P. Beales

Reviewer #3: No

Reviewer #4: No

---

## [Author Response · Author response to Decision Letter 0]

16 Jul 2021

I am sorry that it took a lot of time to revise our manuscript. As you pointed out, the first paper we submitted had many problems with study design, data presentation and data analyses. We have changes substantially of the data presentation and statistical analysis, and corrected according to the indications as much as possible.

---

## [Decision Letter · Decision Letter 1]

24 Aug 2021

PONE-D-20-38582R1

Thrombotic events and rebleeding after hemorrhage in patients taking direct oral anticoagulants

PLOS ONE

Dear Dr. Abe,

Thank you for submitting your manuscript to PLOS ONE. After careful consideration, we feel that it has merit but does not fully meet PLOS ONE’s publication criteria as it currently stands. Therefore, we invite you to submit a revised version of the manuscript that addresses the points raised during the review process.

Your paper was reevaluated by the previous reviewers. Though the manuscripts was improved, minor points still need to be clarified. Please read the comments carefully and address the issues accordingly.

We look forward to receiving your revised manuscript.

Kind regards,

Tomohiko Ai, M.D., Ph.D.

Academic Editor

PLOS ONE

Journal Requirements:

Reviewers' comments:

Reviewer's Responses to Questions

**Comments to the Author**

1. If the authors have adequately addressed your comments raised in a previous round of review and you feel that this manuscript is now acceptable for publication, you may indicate that here to bypass the “Comments to the Author” section, enter your conflict of interest statement in the “Confidential to Editor” section, and submit your "Accept" recommendation.

Reviewer #1: (No Response)

Reviewer #2: All comments have been addressed

Reviewer #3: (No Response)

2. Is the manuscript technically sound, and do the data support the conclusions?

Reviewer #1: Partly

Reviewer #2: Yes

Reviewer #3: Yes

3. Has the statistical analysis been performed appropriately and rigorously? 

Reviewer #1: Yes

Reviewer #2: No

Reviewer #3: I Don't Know

4. Have the authors made all data underlying the findings in their manuscript fully available?

Reviewer #1: Yes

Reviewer #2: Yes

Reviewer #3: Yes

5. Is the manuscript presented in an intelligible fashion and written in standard English?

Reviewer #1: Yes

Reviewer #2: Yes

Reviewer #3: Yes

6. Review Comments to the Author

Reviewer #1: The authors have tackled some of the issues I (and others reviewers have raised). I appreciate their effort and I thank them. I do think this is a more balanced manuscript and important details have been added in response to comments other reviewers have raised. As far as I go, I have a few minor issues and a comment that pertains to the major issue I raised in the previous draft

Major comments

-The authors have removed a comment that spoke to the clinical decisions that physicians should make based on this retrospective study, given the potential biases. However, another such comment is present in the current version. Page 14 line 246 “clinicians should resume anticoagulants after bleeding events….” I still think this study (or any of the other studies published, which are all retrospective and/or observational) cannot be used to reach this conclusion. I suggest removing this statement

Minor concerns/suggestions:

-Most importantly for clarity I think the authors should use consistently bleeding for the first bleeding (the one leading to study entry) and rebleeding for subsequent hemorrhages. While they occasionally do, this is not consistent (e.g., section title “Bleeding and thrombosis after initial bleeding” would be “Rebleeding and thrombosis…”), page 9 line 163 “…11 developed rebleeding”, table 3 (title and table itself) etc . I think this would help follow the text more easily.

-Methods. Page 4 lines 76-78 and page 5 lines 92 -93 are redundant (same exclusion criteria are mentioned)

-Statistics section fail to mention most of the analyses conducted

-Table 1 title is confusing as “baseline” and “at index bleeding” seem contradictory (baseline would generally mean at the time of starting anticoagulation)

-Tables 5 and 6 include “E8” on several occasions in the aOR column. Meaning is unclear. Does that mean 10 raised to the power of 8? This seems unplausible but in any event meaning should be cleared up.

Reviewer #2: The revised manuscript is much improved and presents some more data examining the relative safety of restarting DOACs after a bleeding episode. The overall cohort is small and the data are not really any different from several other reports. The data again emphasize the overall safety of early reintroduction of DOACs in this situation. The size of the cohort makes it difficult to draw strong conclusions on the relative risks.

There are no major sticking points.

Some relatively minor points to consider addressing.

1. Amend the title to include......for non-valvular atrial fibrillation. The study specifically focuses on this group of patients taking DOACs, and although the results are likely to have some generalisability, the risk stratification for other indications for DOACs may be different.

2. The authors give rates of bleeding events and thrombotic events per year, in the Results section. Given the small cohort and limited number of events (especially thrombotic) the confidence intervals around these rates are likely to be high. By just giving a single figure, this suggests that rates of bleeding a nearly three times higher than thromboembolism. I doubt that the data are rich enough to say that for certain. Please include some marker of the range or confidence intervals around these calculations of event rates.

Reviewer #3: the paper is now more informative. My only concern is again related to the use of antiplatelet drugs. It is not clear if rebleeding was mainly recorded in pts who remained on combination therapy. It is unfair to attribute a bleeding (or re-bleeding) on DOAC rather than on antiplatelet if this was also given. At least a comment in the discussion is needed.

7. PLOS authors have the option to publish the peer review history of their article (what does this mean?). If published, this will include your full peer review and any attached files.

Reviewer #1: **Yes: **marc sorigue

Reviewer #2: **Yes: **Ian L. P. Beales

Reviewer #3: No

---

## [Author Response · Author response to Decision Letter 1]

11 Sep 2021

We appreciate your repeated careful guidance. We feel that our manuscript is of high quality owing to your cooperation.

---

## [Decision Letter · Decision Letter 2]

10 Nov 2021

PONE-D-20-38582R2Thrombotic events and rebleeding after hemorrhage in patients taking direct oral anticoagulants for non-valvular atrial fibrillationPLOS ONE

Dear Dr. Abe,

Thank you for submitting your manuscript to PLOS ONE. After careful consideration, we feel that it has merit but does not fully meet PLOS ONE’s publication criteria as it currently stands. Therefore, we invite you to submit a revised version of the manuscript that addresses the points raised during the review process.

Your paper was reevaluated by the previous two reviewers. One reviewer did not respond to my invitation for review but the reviewer seems to be satisfied with your answer and revised manuscript. There is one minor modification suggested by one reviewer this time. I would recommend adding a description to discussion (please see the comment). 

We look forward to receiving your revised manuscript.

Kind regards,

Tomohiko Ai, M.D., Ph.D.

Academic Editor

PLOS ONE

Journal Requirements:

Reviewers' comments:

Reviewer's Responses to Questions

**Comments to the Author**

1. If the authors have adequately addressed your comments raised in a previous round of review and you feel that this manuscript is now acceptable for publication, you may indicate that here to bypass the “Comments to the Author” section, enter your conflict of interest statement in the “Confidential to Editor” section, and submit your "Accept" recommendation.

Reviewer #1: All comments have been addressed

Reviewer #3: All comments have been addressed

2. Is the manuscript technically sound, and do the data support the conclusions?

Reviewer #1: Yes

Reviewer #3: Yes

3. Has the statistical analysis been performed appropriately and rigorously? 

Reviewer #1: Yes

Reviewer #3: I Don't Know

4. Have the authors made all data underlying the findings in their manuscript fully available?

Reviewer #1: Yes

Reviewer #3: Yes

5. Is the manuscript presented in an intelligible fashion and written in standard English?

Reviewer #1: Yes

Reviewer #3: Yes

6. Review Comments to the Author

Reviewer #1: (No Response)

Reviewer #3: I think that a brief comment in the discussion should adress the fact that some pts restarted not only the NOAC but the antiplatelet as well.

7. PLOS authors have the option to publish the peer review history of their article (what does this mean?). If published, this will include your full peer review and any attached files.

Reviewer #1: **Yes: **marc sorigue

Reviewer #3: No

---

## [Author Response · Author response to Decision Letter 2]

12 Nov 2021

I am very grateful for the careful proofreading and the accurate advice.

---

## [Editor Report · Decision Letter 3]

15 Nov 2021

Thrombotic events and rebleeding after hemorrhage in patients taking direct oral anticoagulants for non-valvular atrial fibrillation

PONE-D-20-38582R3

Dear Dr. Abe,

We’re pleased to inform you that your manuscript has been judged scientifically suitable for publication and will be formally accepted for publication once it meets all outstanding technical requirements.

Kind regards,

Tomohiko Ai, M.D., Ph.D.

Academic Editor

PLOS ONE
---

## [Editor Report · Acceptance letter]

17 Nov 2021

PONE-D-20-38582R3 

Thrombotic events and rebleeding after hemorrhage in patients taking direct oral anticoagulants for non-valvular atrial fibrillation 

Dear Dr. Abe:

I'm pleased to inform you that your manuscript has been deemed suitable for publication in PLOS ONE. Congratulations! Your manuscript is now with our production department. 

Kind regards, 

on behalf of

Dr. Tomohiko Ai 

Academic Editor

PLOS ONE